# Recent Advances in the Systemic Treatment of Localized Gastroesophageal Cancer

**DOI:** 10.3390/cancers15061900

**Published:** 2023-03-22

**Authors:** Hannah Christina Puhr, Thorsten J. Reiter, Matthias Preusser, Gerald W. Prager, Aysegül Ilhan-Mutlu

**Affiliations:** Division of Oncology, Department of Medicine I, Medical University of Vienna, Waehringer Guertel 18-20, 1090 Vienna, Austria

**Keywords:** gastric cancer, esophageal cancer, localized

## Abstract

**Simple Summary:**

Gastroesophageal cancer is a devastating disease with dismal survival even in localized settings. Novel therapeutic regimen are underway to improve patient management and outcome. This review aims to demonstrate the advances of immunotherapy and targeted therapies in treatment of localized gastric, gastroesophageal junction and esophageal tumors and gives a short summary on promising ongoing clinical trials.

**Abstract:**

The overall survival expectancy of localized gastroesophageal cancer patients still remains under 5 years despite advances in neoadjuvant and adjuvant treatment strategies in recent years. For almost a decade, immunotherapy has been successfully implemented as a first-line treatment for various oncological diseases in advanced stages. In the case of advanced gastroesophageal cancer, 2021 witnessed several approvals of immune checkpoint inhibitor therapies by different authorities. Although it is still a debate whether this treatment should be restricted to a certain subgroup of patients based on biomarker selection, immunotherapy agents are making remarkable steps in resectable settings as well. The Checkmate-577 study demonstrated significant benefits of nivolumab as an adjuvant treatment for resectable esophageal and gastroesophageal junction tumors and thereby obtained approvals both from U.S. American and European authorities. First results of further potential practice-changing clinical trials are expected in 2023, which might change the treatment armamentarium for resectable gastroesophageal cancers significantly. This review aims to demonstrate the advances of immunotherapy and targeted therapies in treatment of localized gastric, gastroesophageal junction and esophageal tumors and gives a short summary on promising ongoing clinical trials.

## 1. Introduction

Gastroesophageal cancer is a devastating disease with around 1.6 million newly diagnosed cases per year [1]. Although cancer of the upper gastrointestinal tract is often viewed as one entity, there are vast differences in tumor location and histology. Molecular characterization of gastroesophageal cancers even suggests further differentiation, which will play an important role in future trials and patient management [2,3]. Yet so far, clinical practice still focuses on major histological subgroups.

Whereas squamous cell carcinomas (SCC) are commonly found in the esophagus and predominantly seen in patients of Asian ethnicity, carcinomas in the stomach and the gastroesophageal junction (GEJ) are usually adenocarcinomas (AC) and exert the most cases in Western countries. Although etiology, pathobiology and treatment strategies vary depending on the location of the tumor and the histological subtype, a combination of different tumor locations and even histological subgroups is still common practice in clinical cancer trials as it is in everyday patient care [4,5]. As subgroup analyses show tissue agnostic benefit of certain therapeutic strategies, predictive molecular markers gain more focus in trial designs.

However, overall survival (OS) is dismal independent of histology and tumor location, as even in resectable settings, the 5-year survival still only reaches 32–72% in stomach and 26–48% in esophageal cancer patients [6,7]. Thus, novel treatment approaches are warranted to improve patient management. This review aims to provide a concise overview of current and possible future clinical trials in patients with localized gastroesophageal cancer based on tumor location and histological subtype, while putting special emphasis on predictive molecular markers. Since treatment strategies in localized settings can differ significantly depending on the tumor site despite similar histology, this review includes separate sections on gastric adenocarcinoma and adenocarcinoma of the esophageal/gastroesophageal junction to address this issue. Esophageal squamous cell carcinoma is addressed in an additional section to emphasize the importance of histological subtypes in clinical cancer trials as well as in everyday clinical routine.

## 2. Molecular Markers

The last decade has led to a paradigm shift in the treatment of several oncological diseases. The combination of well-established therapeutic approaches such as chemotherapy, radiation therapy and surgery with novel strategies such as targeted therapy and immunotherapy has altered the way of patient management. Especially in advanced stages, these combinations have found their way into clinical routine. Yet, novel treatment agents are currently making remarkable steps in resectable settings as well.

The question remains whether these novel treatment approaches are feasible options for all patients or only small subgroups. Thus, the evaluation of molecular markers in addition to the differentiation between tumor location and SCC versus AC may be crucial.

While the evaluation of human epidermal growth factor receptor 2 (Her2) has been routinely performed in patients with gastroesophageal AC for years and is a feasible surrogate parameter for the response to Her2 blockage in advanced settings [8], the role of Her2 and especially Her2-low expression in localized settings is still evaluated [9]. Although patient selection for targeted treatment still poses an issue for clinical trials, the analysis of Her2 expression remains robust [10].

However, the evaluation of other molecular markers to determine treatment success remains challenging. Immunohistochemical staining to determine programmed death-ligand 1 (PD-L1) expression can be evaluated by several different scores. The most prominent methods are the tumor proportion score (TPS) and the combined positive score (CPS). While the TPS is defined by the percentage of viable tumor cells showing partial or complete membrane staining, the CPS score also includes positive immune cells (lymphocytes and macrophages), adds them to the positive tumor cells and puts them in relation to total tumor cells. Another emerging scoring system represents the so-called “tissue area positivity” (TAP) of PD-L1. TAP has been integrated into clinical trials investigating the efficacy of the anti PD-1 antibody tislelizumab in Asian and Caucasian patients with gastroesophageal cancer [11]. However, not only the scoring system might vary depending on clinical trials, but also the immunohistochemical staining may differ depending on the methods used [12]. This issue has led to a major discussion of whether PD-L1 testing is an adequate predictive marker for immunotherapy. Thus, when interpreting study results, it is crucial to thoroughly asses the methodology. Nevertheless, authorities such as the Food and Drug administration (FDA) or European Medicines Agency (EMA) did not restrict their approval for immunotherapies in gastroesophageal cancer to certain PD-L1 tests, giving the clinicians the freedom to use one of the established antibodies for PD-L1 staining.

Further immune checkpoints which play an important role in the development of gastroesophageal cancers and are currently under investigation within various clinical trials are “cytotoxic T-lymphocyte-associated protein 4” (CTLA4) on regulatory and conventional T cells [13] as well as “T cell immunoreceptor with Ig and ITIM domains” (TIGIT), which is presented on T cells as well as natural killer cells [14].

Other markers, besides immune checkpoints, that might be associated with treatment response to immunotherapy are microsatellite instability high (MSI-H) or mismatch repair deficient (dMMR) and tumor mutational burden high (TMB) tumors, which are already established biomarkers in metastatic settings, as well as infection with Epstein–Barr virus (EBV positive) and specific gene-expression profiles (GEP) [15]. Although none of those biomarkers has been established in localized settings yet, they might pose interesting surrogate parameters in the near future. It is surmised that MSI and EBV positivity is substantially more frequent in localized settings than in advanced cancers. The Cancer Genome Atlas (TCGA) even differentiated gastric cancer into four subgroups—EBV positive, MSI-H, genomically stable and chromosomal instable tumors [2]. In addition, recent analyses show that TMB high tumors might represent their own category [16]. However, these classifications have not found their way into clinical routine yet.

Further markers which might be interesting targets in the near future are claudin18.2 (CLDN18.2), a tight-junction protein, and the Fibroblast Growth Factor Receptor (FGFR) pathway [17,18].

Considering other molecular markers, there are several hereditary gene mutations that are associated with gastroesophageal cancer [19]. However, genetic mutations in gastric cancer are rare (approximately 3%) and, with the exception of mutations in mismatch repair genes leading to mismatch repair deficiency, do not offer any treatment consequence currently.

In addition to histopathological markers, recent studies show that circulating tumor DNA (ctDNA), which is a fraction of tumor DNA that can be detected by a liquid biopsy through a blood draw, might play an important role in predicting disease progression. Although there are currently no recommendations that ctDNA should be evaluated throughout patient history, this marker was shown to predict prognosis as well as tumor relapse in several cancer entities [20]. Thus, ctDNA is an interesting follow-up parameter in ongoing clinical trials in localized settings.

The following sections give an overview of novel systemic treatment approaches based on location and histology of upper gastrointestinal tumors as well as recent approvals of European and U.S. authorities.

## 3. Gastric Adenocarcinoma

Localized gastric AC encompasses stages Ia–III according to the American Joint Committee on Cancer (AJCC/UICC) TNM classification 8th edition. However, stage Ia (T1N0M0) can usually be removed either endoscopically, or if that is not possible, surgically, and no further systemic treatment is recommended. Stages Ib–III (>T1 and/or ≥N0M0) require multimodality treatment including perioperative chemotherapy [4]. For gastric AC, the current established treatment is perioperative chemotherapy with a triplet chemotherapy regimen (fluorouracil plus leucovorin, oxaliplatin and docetaxel). This treatment, called FLOT, has been well established for years; however, as mentioned before, relapses after curative surgery are frequent, and even in the initial FLOT4 trial, the OS was slightly over 4 years [21].

Several ongoing trials aim to further improve survival in curative treated patients with gastric AC and are potentially paradigm changing. Large trials for immunotherapy agents are summarized in Table 1 and Her2-targeted therapies in Table 2.

### 3.1. Immunotherapy

The same group that published the paradigm-changing FLOT-trial (FLOT-AIO German Gastric Cancer Group) recruited 295 patients with resectable gastroesophageal cancer in Germany and Switzerland and randomized them in addition to standard procedure to immunotherapy or placebo. First results of this phase II trial (DANTE) were presented at the annual meeting of the American Society of Clinical Oncology (ASCO) 2022 and showed that the addition of atezolizumab to FLOT (atezolizumab 840 mg every 2 weeks in combination with FLOT followed by atezolizumab monotherapy 1200 mg for 8 cycles every 3 weeks) in comparison to the chemotherapy regimen alone is feasible. Downsizing favored immunochemotherapy over chemotherapy alone (pT0, 23% vs. 15%; pN0, 68% vs. 54%, respectively), and increases in regression grades were also observed. Patients with high PD-L1 expression (CPS ≥ 10) had a particularly favorable response to the combination therapy [22]. Survival analysis will show whether these promising first results can be translated into superior OS. These initial results motivated the design of a following phase III trial examining a similar population; however, a biomarker preselection based on PD-L1 expression is planned.

Another study investigating the combination of FLOT + immunotherapy is the KEYNOTE-585, a double-blind international phase III trial with an estimated sample size of over 1000 patients. As the combination therapy with FLOT is effective, yet toxic, the trial also evaluates the combination with a doublet chemotherapy regimen. Patients are randomized to 3 cycles of pembrolizumab or placebo in combination with either FLOT or cisplatin/fluoropyrimidine before and 3 cycles after surgery and then up to 11 cycles of pembrolizumab or placebo monotherapy [23].

Another international, double-blind, randomized phase III trial is the MATTERHORN trial, which plans to recruit over 900 patients and compares durvalumab with placebo in combination with FLOT [24].

So far, no results of these trials have been reported, and their outcomes are estimated for 2023–2024.

In addition to these trials with triplet chemotherapy backbones, first results of a U.S. trial investigating the combination of pembrolizumab with doublet chemotherapy with capecitabine and oxaliplatin (CAPOX) were presented at the 2022 annual meeting of the American Association for Cancer Research (AACR). In this phase II trial, 36 patients received 3 cycles of CAPOX + pembrolizumab (pembrolizumab 200 mg in combination with CAPOX every 3 weeks) before surgery as well as 3 cycles after surgery and 12 months pembrolizumab maintenance therapy. The primary endpoint could be reached with 20% pathological complete response (pCR) rate, and PFS and OS had not yet been reached [25]. However, grade 3 and greater adverse events were reported in more than half of the cohort, and 3 deaths occurred during the trial, 2 of which were possibly treatment-related. Hence, as doublet chemotherapy is not the current standard of care for fit patients, and major adverse events occurred during this study, further investigation of this regimen is in order before implementing it in clinical routine.

Another approach to immunotherapy in an adjuvant setting was proposed as a phase II trial by the European Organisation for Research and Treatment of Cancer (EORTC). In the VESTIGE trial, 197 European patients who received standard neoadjuvant chemotherapy with surgical resection and showed a high risk of recurrence in pathological work-up (N+ and/or R1 resections) were randomized to receive either standard adjuvant chemotherapy (completion of the perioperative management) or immunotherapy with nivolumab and low-dose ipilimumab (nivolumab 3 mg/kg iv every 2 weeks plus ipilimumab 1 mg/kg iv every 6 weeks for 1 year) [26].

### 3.2. Microsatellite Instability High (MSI-H)/Mismatch Repair Deficient (dMMR) Adenocarcinoma

Patients with MSI-H/dMMR tumors, including patients with hereditary Lynch syndrome, represent a specific subgroup in cancer care, as several studies have shown that they respond exceptionably well to immunotherapy. Thus, MSI status has a major influence on perioperative outcome. Recent advances in other tumor entities might even pose the question of whether surgery has to be performed when complete response after immunotherapy can be achieved [27]. The evaluation of the MSI-H/dMMR gastroesophageal AC subgroup, which is estimated to be up to 15% of all patients, is currently under investigation by several clinical trials.

The interim results of the aforementioned DANTE trial comprised 25 MSI-H patients (8%) and showed exceptionally good tumor regression when treated with FLOT + atezolizumab compared to FLOT monotherapy. The tumor regression grade T1a/b according to Becker (T1a: no residual tumor; T1b: <10% residual tumor [28]) was 70% vs. 47% [22]. Another study investigating immunochemotherapy in MSI-H gastric cancer patients in curative settings is currently performed in China evaluating the adjuvant combination of camrelizumab (PD-1 inhibitor) with docetaxel + S-1 [29]. As the primary endpoint of this analysis is the 3-year disease-free survival rate, results are expected within the next few years.

In addition, several studies evaluated combinations of different checkpoint inhibitors without chemotherapy backbones in MSI-H upper-GI AC patients. In the phase II NEONIPIGA trial, 32 patients received neoadjuvant nivolumab (PD-1 inhibitor; 240 mg once every 2 weeks × 6) and ipilimumab (CTLA-4 inhibitor; 1 mg/kg once every 6 weeks × 2), followed by surgery and adjuvant nivolumab (480 mg once every 4 weeks × 9). Interestingly, 50% of patients had a tumor localized at the gastroesophageal junction, which was surprising, as MSI-H/dMMR AC arise mostly from the body of the stomach. This emphasizes the recommendation that screening for MSI should be performed independent of tumor location [30].

Out of 29 patients who received surgery (91%), 17 (59%) achieved a complete pathological response (T0N0). Three patients (9%) had no visible tumor cells in endoscopy assessment after the neoadjuvant period; two of them refused surgery, and one patient was mistakenly classified as resectable although the patient was initially in a metastatic setting and therefore did not undergo surgery. Other patients had regression grades 1a–3 according to Becker [30]. Although there were no unexpected toxicities, in 19% of patients, grade ¾ adverse events occurred, and one patient died postoperatively (severe cardiovascular comorbidity-related death) [30]. There were no relapses at the time of database lock, thus raising the question of whether surgery may be avoided in this cohort altogether. The majority of patients with MSI-H/dMMR cancer are elderly [2]. Since morbidity after gastrectomy is high, a watch-and-wait strategy might be preferable for the elderly population when achieving pathological complete response (pCR) after neoadjuvant treatment. However, due to an 80 year old patient refusing surgery and the high surgical morbidity and mortality for elderly patients, an amendment was performed to exclude patients older than 75 years from the NEONIPIGA trial. Thus, no further data on elderly patients could be gathered [30].

However, additional studies investigating MSI-H/dMMR upper GI patients are underway and might shed more light on this area. The Italian INFINITY trial is an ongoing phase II, multicenter, single-arm, multi-cohort analysis investigating the activity and safety of tremelimumab (CTLA-4 inhibitor) and durvalumab (PD-L1 antibody) as neoadjuvant (Cohort 1) or potentially definitive (Cohort 2) treatment for MSI-H/dMMR/EBV-negative, resectable gastric and gastroesophageal junction cancer [31]. The primary endpoint of Cohort 1 is the pCR rate as in the NEONIPIGA trial. In parallel to the first reports of the NEONIPIGA trial, the initial results of Cohort 1 of the INFINITY study also show a pCR of around 60%. However, further disease assessment for complete response was planned to be done by ctDNA after neoadjuvant immunotherapy, and all patients with pCR had negative ctDNA pre-surgery [32]. For Cohort 2, the primary endpoint is the 2-year complete response rate, defined as the absence of macroscopic or microscopic residual disease at radiological examinations, tissue and liquid biopsy, during non-operative management without salvage gastrectomy [31]. Thus, the results of this trial might provide evidence on the omission of surgery in specific cancer subgroups.

However, all of these trials are low in patient numbers and have major differences in the scope of study protocols, thus making approval from the authorities based on these results difficult. Larger patient numbers are needed to better understand the role of immunotherapy in this patient cohort. Thus, the IMHOTEP trial evaluates neoadjuvant pembrolizumab in MSI-H/dMMR tumors in a perioperative setting independent of tumor location [33]. This tumor agnostic approach has led to FDA approval in the metastatic setting and might be the key to approval from authorities in a localized setting as well. The trial will include 120 patients and, thus, may have an advantage compared to other studies. As NEONIPIGA and INFINITY trials investigate dual immunotherapy blockade, the question will remain unanswered of whether a single immune checkpoint blockade would be efficient enough to achieve higher response rates in MSI/dMMR patients. The IMHOTEP trial will assess pembrolizumab as a single agent in neoadjuvant phase and hopefully shed some light on the role of mono immune checkpoint inhibitors in this setting.

### 3.3. Human Epidermal Growth Factor Receptor 2 (HER2)

Targeting of human epidermal growth factor receptor 2 (HER2)-positive gastroesophageal tumors has been established in metastatic settings for more than a decade [8]. However, no clear recommendation could be formed for the neoadjuvant setting.

Recently published data from the German Gastric Cancer Study Group showed that the addition of the HER2 antibody trastuzumab to FLOT (loading dose 6 mg/kg, then 4 mg/kg every 2 weeks in combination with FLOT, then 9 cycles of trastuzumab monotherapy) resulted in promising histopathological response rates in 56 patients (pCR 21%, almost pCR 25%) and EFS (median 42.5 months, 3-year survival 82%) without unexpected toxicity [34]. Based on these data, a lot of clinicians treat their fit patients with this combination therapy. However, there is no clear recommendation from international guidelines based on randomized studies and, thus, trastuzumab might not be reimbursed by authorities.

Targeting HER2 beyond trastuzumab has been successfully explored in breast cancer patients, and comparable studies are underway in gastric cancer as well. The PETRARCA trial investigated the addition of trastuzumab and pertuzumab to FLOT (8/6 mg/kg + 840 mg every 3 weeks in combination with FLOT, then 9 cycles of trastuzumab + pertuzumab) versus chemotherapy alone and was planned as a phase II/III study. First results from phase II were promising, with major improvement of pCR rates (12% vs. 35%, *p* = 0.02; respectively) and EFS (median 26 months vs. not yet reached; HR 0.58, *p* = 0.14; respectively) [35]. However, due to the negative JACOB trial, which investigated pertuzumab versus placebo in combination with trastuzumab + cisplatin + fluoropyrimidine in metastatic HER2-positive gastroesophageal tumor patients (median OS 17.5 months (95% CI 16.2–19.3) vs. 14.2 months (12.9–15.5); HR 0.84 (95% CI 0.71–1.00); *p* = 0.057) [36], the PETRARCA trial was closed before phase III initiation.

A promising ongoing trial is the INNOVATION trial organized by the European Organization for Research and Treatment of Cancer (EORTC). This trial has 3 arms: (a) chemotherapy mono (FLOT or oxaliplatin/fluoropyrimidine; every 2 weeks, 4 times before and 4 times after surgery); (b) chemotherapy + trastuzumab (every 3 weeks, 3 times before and 3 times after surgery); (c) chemotherapy + trastuzumab + pertuzumab (every 3 weeks, 3 times before and 3 times after surgery). Adjuvant treatment either with trastuzumab or trastuzumab + pertuzumab is planned to continue for 17 cycles in total [37].

Further approaches to target the HER2 pathway include the elegant concept of antibody–drug conjugates (ADCs). Although cytotoxic agents are components of these novel drugs, unlike historical chemotherapy, ADCs are intended to target and kill tumor cells via a specific receptor while sparing healthy cells. Although there are not yet positive results in localized settings, recent trials in advanced settings in patients pretreated with trastuzumab showed that these armed antibodies are potential weapons against cancer progression. The DESTINY-Gastric 01 study compared trastuzumab–deruxtecan (T-DXd) with standard chemotherapy (irinotecan or paclitaxel) in a third-line setting and showed a clear survival benefit (OS 12.5 vs. 8.4 months; HR 0.59 (95% CI 0.39–0.88), *p* = 0.01) [38], which led to a fast approval of T-DXd by Asian authorities and the FDA [39]. The DESTINY-Gastric 02 and DESTINY-Gastric 04 trials are currently evaluating this treatment in a second-line setting [40]. First results of the DESTINY-Gastric 02 show promising overall response rates of 42% with 4 complete responses. The median survival is also promising with 12.1 months (95% CI 9.4–15.4) [41]. However, it has to be addressed that 50% of patients in this cohort suffered from major adverse events (grade III), and 2 patients even died due to therapy-associated pneumonitis. Thus, although this novel therapeutic strategy opens new doors to cancer management, constant vigilance is essential. Based on these data, T-DXd was approved by the EMA for the treatment of second-line Her2-positive gastric cancer pretreated with trastuzumab.

The near future will bring trials investigating ADCs in curative settings. Currently, the phase II EPOC2003 trial is recruiting HER2-positive patients in Japan to receive 3 cycles of TDXd neoadjuvantly [42]. Major pathological response (MPR) was chosen as the primary endpoint and is defined as the proportion of subjects with <10% residual tumor in the stomach and lymph nodes by central assessment. However, whether TDXd is used adjuvantly in patients with excellent response is not specified in the published materials so far, and results from this highly innovative trial are expected for 2023–2024.

Another approach is the combination of HER2-targeted therapy and immunotherapy. The KEYNOTE-811 trial investigates the combination of trastuzumab + chemotherapy + either pembrolizumab or placebo in first-line advanced gastroesophageal tumors and showed promising overall response rates of 74.4% (95% CI 66.2–81.6) vs. 51.9% (95% CI 43.0–60.7; *p* = 0.00006) respectively, thereby gaining an approval by the FDA [43]. Due to the increased response rates, this concept needs to be evaluated in the neoadjuvant setting. Thus, the German AIO study group has proposed a phase II trial investigating the combination of pembrolizumab, trastuzumab and FLOT as perioperative treatment of HER2-positive, localized esophagogastric adenocarcinoma. The trial is planned to start in 2023 and will investigate the pCR and 2-year DFS in 30 patients. Results of this trial are expected for 2025 [44].

Although neither ADCs nor Her2-PD-(L)1-combination therapies are yet approved in localized settings, future study protocols will provide more insight in this matter and potentially change the face of how to treat localized HER2-positive disease.

### 3.4. Antiangiogenic Agents

Another approach to target tumor cells lies in antiangiogenic agents, which prevent the tumor from spreading new vessels and, thus, reduce cancer growth. One of the most widely used antiangiogenic drugs is the monoclonal antibody bevacizumab, which inhibits vascular endothelial growth factor A (VEGF-A) and slows down the growth of blood vessels. Although this targeted therapy showed immense potential in other cancer entities such as colorectal or lung cancer and also non-oncological diseases such as macular degeneration, this effect could not be transferred to gastroesophageal AC patients, neither in advanced nor in localized settings. The British ST03 trial enrolled over 1000 patients with resectable gastroesophageal AC between 2007 and 2014 and randomized them to receive either 3 cycles pre- and 3 cycles postoperative chemotherapy alone (then standard care: epirubicin, cisplatin and capecitabine) or chemotherapy in combination with bevacizumab 7.5 mg/kg every 3 weeks. Unfortunately, no improvement of the OS was observed after 3 years (50.3% (95% CI 45.5–54.9) vs. 48.1% (43.2–52.7); HR 1.08 (95% CI 0.91–1.29; *p* = 0.36)). However, in patients receiving bevacizumab, wound-healing complications (*n* = 33 (7%) vs. *n* = 53 (12%)) and post-operative anastomotic leaks (*n* = 23 (10%) vs. *n* = 52 (24%)) were more frequent, showing an unfavorable toxicity profile [45].

Another approach to target angiogenesis was done in the scope of the RAMSES trial, which evaluated the VEGF-inhibitor ramucirumab in combination with FLOT. As ramucirumab alone or in combination with taxanes showed promising effects in metastatic patients and represents the standard of care as a second-line treatment, expectations were high in resectable settings as well [4,46,47]. Final results were presented at the ASCO 2022 meeting. Patients (*n* = 180) in the randomized phase II/III trial were allocated to receive either perioperative FLOT or FLOT + ramucirumab every 2 weeks followed by 16 cycles of ramucirumab monotherapy. It is important to note that the cohort has a high number of diffuse carcinomas with signet-ring cell components of 45%. It is surmised that signet-ring cell AC have a worse outcome due to reduced chemosensitivity in advanced settings [48]. Unfortunately, the outcome was not improved by addition of ramucirumab, with a very discouraging median OS of 46 vs. 45 months. However, it has to be mentioned that although baseline characteristics were similar between both arms, the study cohort was still not well balanced, with more unfavorable patients in the combination therapy arm (T4 (8% vs. 5%), impaired ECOG PS of 1 (32% vs. 20%) and concomitant disease (86% vs. 76%)). Although R0-resection rates were higher with the addition of the VEGF-inhibitor (96% versus 82%; *p* = 0.0093), surgical morbidity (41% vs. 32%) and rate of grade ≥3 adverse events (92% versus 76%) were higher [49].

In summary, neither bevacizumab nor ramucirumab could meet the high expectations that were put on them based on other tumor entities or metastatic settings. As other targeted therapies are currently on the rise, there are currently no mentionable novel approaches concerning antiangiogenesis. However, a combination of antiangiogenic compounds with immunotherapy agents may still be a reasonable approach, which might be tackled in the near future.

### 3.5. Other Targets

Several other targeted therapies are currently under investigation. The most promising studies target the tight-junction protein CLDN18.2, which is mainly expressed in normal gastric mucosa but becomes exposed upon malignant transformation, or the FGFR pathway, which plays an important role in tissue repair and mutations associated with cancer development.

Zolbetuximab mediates specific killing of CLDN18.2-positive cells through immune effector mechanisms and showed extraordinary efficacy (HR 0.55; 95% CI 0.39–0.77; *p* < 0.0005) in combination with chemotherapy in patients with advanced gastroesophageal AC with a moderate-to-strong expression of CLDN18.2 (≥40% of tumor cells) [50]. However, which patient cohort profits the most from this novel treatment has yet to be determined. Based on the results of the phase II FAST trial, two phase III trials evaluating zolbetuximab in advanced gastroesophageal AC are recruiting to gather further information and determine the optimal chemotherapy backbone. The SPOTLIGHT and the GLOW trial randomized patients when ≥75% of tumor cells showed moderate to strong membranous immunohistochemical staining [51,52]. First results of the SPOTLIGHT trial were presented at the 2023 ASCO GI Cancers Symposium and showed superior PFS (10.61 vs. 8.67 months; HR 0.751; 95% CI 0.589–0.942; *p* = 0.0066) as well as superior OS (18.23 vs. 15.54 months; HR 0.75; 95% CI 0.601–0.936); *p* = 0.0053) for FOLFOX + zolbetuximab in comparison to FOLFOX + placebo in advanced and metastatic patients [53]. These results represent a major breakthrough in targeted therapies for gastroesophageal cancer and may lead to further trials in resectable settings.

The findings of the phase II FIGHT study comparing the efficacy of bemarituzumab, an FGFR2b inhibitor, plus chemotherapy in comparison to placebo plus chemotherapy, also showed promising results in advanced settings. Median PFS as the primary endpoint was 9.5 months (95% CI 7.3–12.9) in the bemarituzumab group and 7.4 months (5.8–8.4) in the placebo group (HR 0.68; 95% CI 0.44–1.04; *p* = 0.073). In addition, median OS as a secondary endpoint was not reached for the bemarituzumab group (95% CI 13.8 to not reached) versus 12.9 months (95% CI 9.1–15.0) in the placebo group (HR 0.58; 95% CI 0.35–0.95); *p* = 0.027) [54]. The antibody was well tolerated by most of the patients and in general showed a favorable toxicity profile compared to the placebo. However, 67% of the bemarituzumab cohort suffered from corneal events (adverse events of special interest; compared to 10% in the placebo cohort), and in 3 patients (4%), treatment-related deaths occurred (compared to 0% in placebo cohort). Although not statistically significant, the prolonging of PFS may be interpreted as a first success and, thus, two further studies (FORTITUDE-101 AND FORTITUDE-102) are recruiting patients with advanced gastroesophageal cancer to evaluate efficacy and safety [55,56].

Although there are currently no studies in localized settings, it might be only a matter of time before these targeted therapies are evaluated perioperatively.

## 4. Esophageal and Gastroesophageal Junction Adenocarcinoma

For the correct staging of esophageal and gastroesophageal junction (GEJ) AC, tumor location has to be considered. Based on the Siewert classification, tumors can be categorized in 3 groups: type I tumors (adenocarcinoma of the distal esophagus), type II tumors (true carcinoma of the cardia) and type III tumors (subcardial gastric cancer infiltrating the distal esophagus) [57]. Siewert type III GEJ AC is staged and treated according to the AJCC/UICC TNM classification 8th edition for gastric cancer and gastric cancer guidelines [4]. Yet, esophageal as well as Siewert type I and II ACs are staged by the AJCC/UICC TNM classification for esophageal cancer. Early disease (T1N0M0) may be managed with endoscopic, or if not possible, surgical resection, and locally advanced tumors (T2–T4 or N1–N3) can either be treated with perioperative chemotherapy or with neoadjuvant radiochemotherapy [5].

The results of the CROSS trial showed that radiochemotherapy before surgical resection provides a feasible therapeutic option for patients with esophageal or GEJ AC [58,59]. However, the 10-year OS data of the CROSS trial stated that for patients with SCC, the profit of radiochemotherapy seems to be greater than for AC (10-year OS comparing surgery only with perioperative CROSS + surgery: SCC (23% of study population) 23% vs. 46% (*p* = 0.007) and AC (75% of study population) 26% vs. 36% (*p* = 0.061)) [60].

Thus, several studies investigated which of the two therapeutic options is more feasible for the patient cohort with AC. The Neo-AEGIS trial randomized 377 European patients to either CROSS or perioperative chemotherapy. First OS analyses showed no statistically significant difference in the 3-year OS (HR 1.02; 95% CI 0.74–1.42). However, results of the trial have to be viewed critically, as most of the evaluable perioperative patients were not treated with the current standard of care FLOT (ECF/ECX/EOF/EOX pre-2018, FLOT option 2019/20, only 15% of the patients in chemotherapy arm received FLOT) [61]. Thus, another trial (ESOPEC) was established to compare the efficacy of neoadjuvant chemoradiation (CROSS) followed by surgery versus perioperative chemotherapy (FLOT protocol) and surgery for the curative treatment of localized esophageal AC [62]. Results are awaited to shed some light on how to best treat this patient subgroup. However, there are also other efforts to optimize treatment options for this patient cohort.

### 4.1. Immunotherapy

Evidence on the addition of targeted- or immunotherapy in esophageal AC is scarce. Patients with GEJ AC are usually included in trials concerning gastric cancer and can be found in the sections above.

The novel standard of care for patients with esophageal and GEJ cancer without complete response after perioperative chemotherapy and R0 resection is adjuvant nivolumab for 1 year according to the CheckMate-577 trial. The trial investigated AC as well as SCC patients and found significant benefits for the overall population (22.4 (16.6–34.0) vs. 11.0 (8.3–14.3) months; HR 0.69; 96.4% CI 0.56–0.86; *p* < 0.001). Although most esophageal cancers are SCC, the CheckMate-577 trial comprised 71% AC patients. Again, the benefit of adjuvant immunotherapy for AC patients was not as pronounced as in SCC patients; however, there is still a statistically significant and clinically meaningful benefit (DFS 19.4 months (95%CI 15.9–29.4) vs. 11.1 (95% CI 8.3–16.8) months; HR 0.75; 95% CI 0.59–0.96) [63].

### 4.2. Targeted Therapy

In another attempt to further improve neoadjuvant radiochemotherapy for this patient cohort, the RTOG-1010 trial investigated whether the addition of trastuzumab to CROSS (4 mg/kg in week one, 2 mg/kg per week for 5 weeks during chemoradiotherapy, 6 mg/kg once pre-surgery, and 6 mg/kg every 3 weeks for 13 treatments starting 21–56 days after surgery) can improve the outcome of HER2-positive GEJ AC patients. Unfortunately, the primary endpoint DFS was not improved by the additional targeted therapy (median DFS 19.6 months (95% CI 13.5–26.2) with chemoradiotherapy plus trastuzumab vs. 14.2 months (10.5–23.0) for chemoradiotherapy alone (HR 0.99; 95% CI 0.71–1.39); log-rank *p* = 0.97) [64].

## 5. Esophageal Squamous Cell Carcinoma

Esophageal and gastroesophageal junction (GEJ) SCC in locally advanced stages (T2–4 or N1–3 with M0 based on the AJCC/UICC 8th edition for esophageal cancer) is treated with either neoadjuvant radiochemotherapy and resection or definitive radiochemotherapy depending on tumor invasion as well as tumor location [5]. Although regimens of radiochemotherapy may change depending on trial data and country, the most commonly used regimen in Europe is based on the CROSS trial (carboplatin + paclitaxel + radiation therapy with 41.4 Gy in 23 fractions) [58,59]. This treatment strategy has been established for a decade, and recently published data show that especially patients with SCC (*n* = 84; 23% of study population) profit from the addition of radiochemotherapy compared to surgery alone (10-year OS: 46% vs. 23%, *p* = 0.007; respectively) [60]. Yet recurrence rates after this standard of care treatment are still high, and further improvement is warranted.

### 5.1. Immunotherapy

As other SCC entities such as lung or head-and-neck cancer are known to respond well to immunotherapy, this approach has been the main focus of several clinical trials including esophageal SCC (ESCC). So far, various immunotherapy compounds were shown to be beneficial in metastatic setting, and one approach led to practice-changing results in resectable setting.

The addition of adjuvant nivolumab after radiochemotherapy and surgical R0 resection without pCR was evaluated in the CheckMate 577 trial, which was already mentioned above. In 230 patients with ESCC and R0 resections without complete pathological response after radiochemotherapy adjuvant immunotherapy for 1 year, the disease-free survival (DFS) more than doubled compared to the placebo (29.7 months (95% CI 14.4–not reached) vs. 11.0 (7.6–17.8); HR 0.61; 95% CI 0.42–0.88) [63]. As this study was conducted independent of PD-L1 expression, the FDA and EMA approved this treatment biomarker independent for patients with residual pathological disease after radiochemotherapy and surgical resection. However, it is important to mention that the post-hoc analysis results found the greatest benefit in patients with a baseline CPS ≥ 5.

Other studies investigating immunotherapy in a perioperative setting are scarce in European cohorts, but there are several ongoing trials in Asia, which are listed in Table 3. However, the current standard treatment strategies in these cohorts might differ significantly from European standards and should therefore not be translated to Western populations.

Whether patients without tumor resection also profit from adjuvant immunotherapy is currently under investigation. The SKYSCRAPER-07 trial will investigate immunotherapy with atezolizumab as well as double-inhibition by atezolizumab plus tiragolumab, a TIGIT antibody, or placebo or double-placebo in unresectable esophageal SCC patients, whose cancers have not progressed following definitive radiochemotherapy [65]. The rationale for this combination therapy is based on the results of the phase II CITYSCAPE trial [66], which showed a clinically meaningful improvement in objective response rate and progression-free survival of the double-immunotherapy in patients with metastatic non-small-cell lung cancer.

Furthermore, the KEYNOTE-975 randomized placebo-controlled study investigates the checkpoint inhibitor pembrolizumab in combination with definitive radiochemotherapy. In contrast to the SKYSCRAPER-07 trial, the administration of immunotherapy will start before initiation of radiochemotherapy and will be continued for up to one year. In addition, this study investigates several chemotherapy backbones [67].

### 5.2. Targeted Therapy

Targeted therapy approaches in localized esophagogastric SCC patients are scarce. However, recent interim results from the Chinese NXCELL1311 trial presented at ASCO 2022 promises efficacy of nimotuzumab, an anti-epidermal growth factor receptor (EGFR) humanized monoclonal antibody. Combined with concurrent CROSS radiochemotherapy, nimotuzumab showed better overall response rates compared to placebo (93.8% vs. 72.0%; *p* < 0.001) [68]. Thus, data on the survival outcome are awaited and might change the perspective on targeted therapy in this patient cohort.

## 6. Discussion

Recent years have changed the face of modern oncology. Approval of immunotherapy in an adjuvant setting after radiochemotherapy and surgery without complete pathological response represents the cornerstone of a new era in the treatment of gastroesophageal cancer. Although this treatment is the new standard approach for both esophageal and gastroesophageal junction tumors independent of histological subtype, in daily clinical practice this therapeutic strategy is only available for a minority of gastroesophageal cancer patients, and vast differences between tumor locations as well as histological and molecular subtypes must be addressed.

The prevalence of AC exceeds SCC in Western countries and rises globally. At the same time, perioperative chemotherapy is often preferred to radiochemotherapy in AC patients, as results from the FLOT trial seem to be more promising compared to CROSS and there are no results from adequate head-to-head comparisons yet. Consequently, the majority of patients still receive chemotherapy regimens without targeted antibodies. Thus, trials to evaluate novel treatment strategies are numerous to improve patient management, and especially in AC, targeted therapy as well as immunotherapy are investigated intensively. However, a multiplicity of immunotherapy studies does not include patients based on molecular profile, which often leads to heterogeneous data. Thus, the evaluation and implementation of molecular biomarkers will potentially be one of the main focuses of novel trial designs. This can already be seen in trials for specific subgroups, i.e. MSI-H/dMMR populations, as well as preliminary patient selection as planned in the phase III trial following results of the DANTE trial favoring a PD-L1 CPS ≥ 10 population.

In particular, HER2 is one of the most intensively studied molecular biomarkers in gastroesophageal tumors. Although promising pathologic response rates and survival outcomes are reported within resectable settings with dual anti-HER2 blockade, no clear recommendation for the implementation of targeted anti-HER2 therapy can be made at this time. Hopefully, further emerging phase II trials and studies investigating novel anti-HER2 targeted modalities, including antibody-drug conjugates, will lead to specific recommendations for the treatment of HER2-positive localized gastroesophageal tumors.

However, it has to be noted that although biomarker-based preselection may lead to more favorable responses, there remains a large number of biomarker-negative patients not profiting from targeted- and immunotherapy. Overcoming these so-called “cold” tumors without favorable biomarkers remains a major challenge. First approaches to sensitize these cold tumors to immunotherapy by evoking dMMR are currently investigated by a phase II trial in advanced settings [69]. The ELEVATE trial investigates the combination of temozolomide and nivolumab in O6-methylguanine-DNA-methyltransferase (MGMT) methylated advanced gastroesophageal cancer patients and will show whether this hypothesis is feasible. Other approaches to overcome immune-cold tumors are sensitizing tumor tissue with radiation therapy. This hypothesis is strengthened by the results of the CheckMate 577 trial, which showed a significant survival benefit in the overall cohort independent of PD-L1 status. The combination of anti-angiogenic or targeted therapy drugs with immunotherapy compounds may again enhance the susceptibility of immune cold tumors to immunotherapy, as seen in KEYNOTE-811 trial investigating the combination of trastuzumab and pembrolizumab [43]. Novel therapeutic strategies might also try to induce response by increasing the frequency of tumor-specific T cells with personalized approaches such as CAR T cell therapy [70]. Future years might shed some light on optimizing treatment strategies for biomarker-negative patient cohorts.

Another issue which has to be addressed are the differences between Asian and Western trials. Trial data for these ethnicities cannot be translated to one another easily. Asian descent is often associated with distinct characteristics of tumor presentation and more favorable outcomes after curative surgery, although the reasons behind this phenomenon are unclear [71,72]. Furthermore, standard treatment strategies differ between Western and Asian countries. Thus, international studies which provide standard-of-care treatment in the control arm are difficult to perform, and therapeutic backbones may reduce feasibility of results for other authorities.

Even in the same geographical regions, several treatment approaches for the same setting are approved by authorities and used by clinicians. As head-to-head comparisons of approved therapeutic regimens are scarce, therapeutic backbones in clinical trials for the same patient population might also differ.

In addition, tumors located in the esophagus and gastroesophageal junction have two major histologies. The differences between AC and SCC have direct clinical impacts. Increasing evidence suggests that patients with SCC in particular are more susceptible to radiation and immunotherapy. However, patients with different histologies have been mixed in clinical trials due to the same anatomical location. Although these studies usually define histological subtype as a stratification factor, the patient population is mostly heterogeneous, making interpretation of the study results difficult. Better patient selection and restriction of inclusion to one histology in future clinical trials will help to generate definite results.

Furthermore, immunotherapeutic and targeted treatments are often continued for several months after surgery. Although immunotherapy is well tolerated, it still can cause severe adverse events as well as an enormous financial burden. Whether these extensive treatment intervals are necessary must also be addressed in future trials. Although there are numerous new approaches to improve gastroesophageal cancer patient management, most of them are still in an early phase of treatment development and further data are needed to change current practices.

## 7. Conclusions

Novel systemic treatment approaches to improve the outcome of patients with localized gastroesophageal cancer are underway. However, heterogenous cohorts might lead to misinterpretation of study results. Thus, the evaluation and implementation of molecular biomarkers, rather than tumor location, plays an important role in future clinical trial designs. Therapeutic regimen, which have been proven to be efficient in metastatic settings, are predisposed to be tested in localized settings soon. Combination of recently discovered treatment strategies, such as immunotherapy and targeted therapy, with established chemotherapeutic regimen as well as ADCs and checkpoint inhibitor monotherapies are currently under investigation in patients with gastroesophageal cancer. In the next few years trial results will potentially open up new paths for the management of gastroesophageal cancer patients.

## Figures and Tables

**Table 1 cancers-15-01900-t001:** Clinical trials investigating immunotherapeutic agents in localized gastroesophageal adenocarcinoma.

Trial Name	National Clinical Trial Number	Phase	Immunotherapy	CHT-Backbone	*N*	Setting	Ethnicity	Primary Endpoint	Recruitment Status
DANTE	NCT03421288	II	Atezolizumab	FLOT	295	Peri-OP	Caucasian	PFS/DFS	Active, not recruiting
KEYNOTE-585	NCT03221426	III	Pembrolizumab	FLOT/CP/FP	800	Peri-OP	Caucasian/Asian	OS/EFS/pCR	Active, not recruiting
MATTERHORN	NCT04592913	III	Durvalumab	FLOT	900	Peri-OP	Caucasian/Asian	EFS	Active, not recruiting
n.a.	NCT02918162	II	Pembrolizumab	CP	36	Peri-OP	Caucasian	pCR	Completed
EORTC-VESTIGE	NCT03443856	II	Nivolumab/Ipilimumab	mono	240	Adjuvant	Caucasian	DFS	Active, not recruiting
ATTRACTION-05	NCT03006705	III	Nivolumab	S1/CP	700	Adjuvant	Asian	RFS	Active, not recruiting
NEONIPIGA	NCT04006262	II	Nivolumab/Ipilimumab	mono	32	Peri-OP	Caucasian	pCR	Recruiting
INFINITY	NCT04817826	II	Durvalumab/Tremelimumab	mono	31	Neoadj/Def	Caucasian	pCR/CRR	Recruiting
IMHOTEP	NCT04795661	II	Pembrolizumab	mono	120	Peri-OP	Caucasian	pCR	Recruiting

Abbreviations: *N* = number of patients; FP = fluoropyrimidine + platin; CP = capecitabine + platin; peri-OP = perioperative; neoadj = neoadjuvant; def = definitive; pCR = pathological complete remission; PFS = progression-free survival; DFS = disease-free survival; OS = overall survival, EFS = event-free survival; RFS = recurrence-free survival; CRR = complete response rate, n.a. = not available. Recruitment status according to clinicaltrials.gov (accessed on 13 December 2022).

**Table 2 cancers-15-01900-t002:** Clinical trials targeting Her2 in localized gastroesophageal adenocarcinoma.

Trial Name	National Clinical Trial Number	Phase	Targeted Therapy	CHT-Backbone	*N*	Setting	Ethnicity	Primary Endpoint	Recruitment Status
HerFLOT	NCT01472029	II	Trastuzumab	FLOT	56	Peri-OP	Caucasian	pCR	Completed
PETRARCA	NCT02581462	II	Trastuzumab/Pertuzumab	FLOT	81	Peri-OP	Caucasian	pCR/PFS	Completed
INNOVATION	NCT02205047	II	Trastuzumab or Trastuzumab/Pertuzumab	FLOT/FP/CP	52	Peri-OP	Caucasian/Asian	pCR	Active, not recruiting
EPOC2003	NCT05034887	II	Trastuzumab-deruxtecan	n.a.	37	Neoadjuvant	Asian	MPR	Recruiting
PHERFLOT	NCT05504720	II	Pembrolizumab/Trastuzumab	FLOT	30	Peri-OP	Caucasian	pCR, DFS	Not yet recruiting

Abbreviations: *N* = number of patients; FP = fluoropyrimidine + platin; CP = capecitabine + platin; pCR = pathological complete remission; PFS = progression-free survival; MPR = major pathological response; peri-OP = perioperative; DFS = disease-free survival; n.a. = not applicable. Recruitment status according to clinicaltrials.gov (accessed on 13 December 2022).

**Table 3 cancers-15-01900-t003:** Immunotherapy trials for patients with localized esophageal squamous cell carcinoma in perioperative settings.

National Clinical Trial Number	Phase	*N*	Immunotherapeutic Agent	Strategy Experimental Arm	Strategy Control Arm	Ethnicity	Primary Endpoint
NCT04807673	III	342	Pembrolizumab (P)	P + CHT + OP + P	CRT + OP	Asian	EFS
NCT05244798	III	360	Tislelizumab (T)	T + CRT + OP	T + CHT + OP	Asian	pCR
NCT04280822	III	400	Toripalimab (To)	To + CHT + OP + To	CHT + OP	Asian	EFS
NCT04973306	II/III	176	Tislelizumab (T)	T + CROSS + OP	CROSS + OP	Asian	pCR
NCT05213312	II/III	90	Nivolumab (N)	N + CHT + OP + (ad N for non-pCR)	CHT + OP + (ad N for non-pCR)	Asian	pCR
NCT05357846	III	422	Sintilimab (S)	S + CRT + OP	CRT + OP	Asian	OS

Abbreviations: *N* = number of patients; CHT = chemotherapy; CRT = chemoradiotherapy; OP = surgery; pCR = pathological complete remission; OS = overall survival, EFS = event-free survival.

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
