# Peer review of "Recent Advances in the Systemic Treatment of Localized Gastroesophageal Cancer"

_cancers, 2023, doi:10.3390/cancers15061900_

Round 1

Reviewer 1 Report

This is a well written summary of the trial situation in resectable upper GI cancer focussing on immunotherapy and some  further molecular targets. The review also points out some critical aspects in the respective clinical trials, without going to much into detail.

Comments:

The combination of targeting different molecular targets such as HER2 and PDL1 has demonstrated encouraging results in advanced upper GI cancer. Some trials evaluating these combinations are on the way and should be mentioned (e.g. PHERFLOT)

Author Response

Thank you for handling our manuscript in such a short time. Please find our responses to your comments attached.

Reviewer 2 Report

The title suggests focus on localized cancer but half of the text is devoted to metastatic cancers

Poorly written and can use considerable editing. 

Poorly organized. needs to re-written entirely. Address: staging, phenotypes, genotypes, immunotypes, treatment based on stages (for example, stage Ia can be treated endoscopically and ib can be treated surgically, etc.)

Need to discuss various guidelines (NA, EU and Asia). discuss differences.

Discuss outcomes in various regions and explain why

Discuss staging approaches in different regions and explain why

Leave out squamous cell carcinoma, it simply confuses the entire issue

Discuss future possibilites

Discuss potential of tissue based and blood based assays

discuss hereditary diffuse type gastric cancer systematically. discuss new guidelines. 

Discuss other germline predisposing factors other than CDH1

Leave out all discussion regarding the advanced stage (which is also done inadequately)

No need to emphasize various PDL-1 testing methods. Why? 

Delete misstatements about PDL-1 testing and regulatory approvals

Remove Claudin18.2 discussion regarding advanced stage. not relevant

TDXd discussion also irrelevant

KN811 irrelevant

The manuscript provides no vision or novel ideas. Reads like a book chapter that is not well organized. 

Author Response

(The authors gave the same response as above.)

Reviewer 3 Report

This is a well-constructed and thorough review of the current status of treatment of localized gastroesophageal cancer.

I have some minor suggestions for the manuscript, as follows:

Introduction

- It might be worth commenting on molecular characterisation of AC and SCC as separate pathologies (eg TGCA data)

Molecular Markers

- Sensible explanations, probably some points need a reference eg. use of HER2. References for CTLA4 and TIGIT.  

- MSI status also has an influence on perioperative chemo outcome

- Claudin 18.2 – Spotlight trial reported +ve => already a target (addressed later in manuscript)

Gastric cancer

- Tables 1+2 – may be helpful to the reader to add a column about reported results vs trial still ongoing

- Destiny Gastric 04 is open

- There are some other novel approaches in advanced setting eg. ELEVATE Trial

Esophageal and GEJ Cancer

- The content is fine, but would it make more sense to group the paper by adenocarcinoma vs squamous, rather than by anatomy? Eg. Gastro-esophageal adenocarcinoma vs. Esophageal squamous carcinoma

Conclusion

- I would suggest expanding the discussion of the importance of the molecular subsets and histology.

Final general comment: Although the manuscript is clear and understandable, there are minor errors of grammar and word choice throughout the manuscript that are noticeable to a first-language English speaker. The authors may wish to address this, though it does not impact on the academic value of the work.

Author Response

(The authors gave the same response as above.)
